# Spin-controlled generation of indistinguishable and distinguishable photons from silicon vacancy centres in silicon carbide

Naoya Morioka[1,2 ✉], Charles Babin[1], Roland Nagy[1], Izel Gediz[1], Erik Hesselmeier[1], Di Liu[1], Matthew Joliffe[1], Matthias Niethammer[1], Durga Dasari[1], Vadim Vorobyov[1], Roman Kolesov[1], Rainer Stöhr[1], Jawad Ul-Hassan[3], Nguyen Tien Son[3], Takeshi Ohshima[4], Péter Udvarhelyi[5,6,7], Gergő Thiering[6], Adam Gali[6,7], Jörg Wrachtrup[1] & Florian Kaiser[1 ✉]

Quantum systems combining indistinguishable photon generation and spin-based quantum information processing are essential for remote quantum applications and networking. However, identification of suitable systems in scalable platforms remains a challenge. Here, we investigate the silicon vacancy centre in silicon carbide and demonstrate controlled emission of indistinguishable and distinguishable photons via coherent spin manipulation. Using strong off-resonant excitation and collecting zero-phonon line photons, we show a two-photon interference contrast close to 90% in Hong-Ou-Mandel type experiments. Further, we exploit the system's intimate spin-photon relation to spin-control the colour and indistinguishability of consecutively emitted photons. Our results provide a deep insight into the system's spin-phonon-photon physics and underline the potential of the industrially compatible silicon carbide platform for measurement-based entanglement distribution and photonic cluster state generation. Additional coupling to quantum registers based on individual nuclear spins would further allow for high-level network-relevant quantum information processing, such as error correction and entanglement purification.

[1] 3rd Institute of Physics, University of Stuttgart and Institute for Quantum Science and Technology IQST, 70569 Stuttgart, Germany. [2] Advanced Research and Innovation Center, DENSO CORPORATION, Nisshin 470-0111, Japan. [3] Department of Physics, Chemistry and Biology, Linköping University, SE-58183 Linköping, Sweden. [4] National Institutes for Quantum and Radiological Science and Technology, Takasaki 370-1292, Japan. [5] Department of Biological Physics, Eötvös University, Pázmány Péter sétány 1/A, 1117 Budapest, Hungary. [6] Wigner Research Centre for Physics, P.O. Box 49, 1525 Budapest, Hungary. [7] Department of Atomic Physics, Budapest University of Technology and Economics, Budafoki út 8., 1111 Budapest, Hungary. ✉email: n.morioka@pi3.uni-stuttgart.de; f.kaiser@pi3.uni-stuttgart.de

The rise of quantum networks depends crucially on scalable quantum systems that combine quantum memories with long coherence times[1,2] and stable optical emission to mediate entanglement via interference of indistinguishable photons[3–8]. In this regard, solid-state quantum systems are very appealing thanks to well-established fabrication techniques that promise chip-based integration and mass production[9]. A caveat coming along with many solid-state systems is strong coupling to charge fluctuations, which leads to large spectral diffusion and ionisation, especially under off-resonant laser excitation and in surface proximity[10,11]. Recently, spectral stability has been greatly improved for divacancy centres in charge-depleted semiconductor diode structures, albeit ionisation remains an issue[12]. Alternatively, one can resort to systems with inversion symmetry, e.g. with germanium-vacancy[13,14], tin-vacancy centres[15], and silicon-vacancy[16] centres in diamond. For the latter, practical multi-millisecond spin coherence times have been demonstrated at ultracold temperatures in the millikelvin range[17].

Here, we use semiconductor colour centres that provide a naturally stable spin–photonic interface at temperatures up to 6.9 K and demonstrate spin-controlled quantum optical interference of indistinguishable and distinguishable photons. This milestone experiment demonstrates that the system possesses the necessary prerequisites for remote entanglement generation as targeted in scalable quantum networks.

Our technological platform is silicon carbide (SiC), which has recently acquired great interest in the quantum community as it is industrially compatible and hosts several appealing optically addressable spin-active quantum defects[18–21]. We investigate here the negatively charged silicon vacancy centre at hexagonal lattice site (h-$V_{Si}$) in the common 4H polytype of SiC[22]. Despite the lack of inversion symmetry, it was recently shown that the optical transition of h-$V_{Si}$ is decoupled from charge fluctuations through identical wavefunction symmetries in the ground and V1 excited states (ESs)[23]. Thanks to weak phonon coupling in the ground state (GS), millisecond spin coherence times are achieved[20,24].

## Results

**Spin-optical system of silicon vacancies in silicon carbide**. Our 4H-SiC host crystal is an isotopically purified (0001) epitaxial layer ($^{28}$Si ~99.85%, $^{12}$C ~99.98%), which is irradiated with 2 MeV electrons to generate isolated single h-$V_{Si}$ centres as shown in Fig. 1a. The crystal is slightly n-type to ensure that h-$V_{Si}$ centres are in the desired negatively charged state (see "Methods"). As shown in Fig. 1b, the h-$V_{Si}$ centre presents a spin quartet system ($S = \frac{3}{2}$). At zero external magnetic field, spin sublevels $m_S = \pm\frac{1}{2}$ and $m_S = \pm\frac{3}{2}$ are pairwise degenerate in the GS and the V1 ES. As the GS zero field splitting is relatively small at 4.5 MHz[22], we apply a magnetic field ($B \approx 9$ G) along the system's symmetry axis (c-axis). This lifts the degeneracy and suppresses parasitic spin mixing due to external stray fields. Optical transitions are linearly polarised and associated with two zero phonon line (ZPL) transitions at 861.7 nm. They are assigned to $|\pm 1/2\rangle_{GS} \leftrightarrow |\pm 1/2\rangle_{ES}$ and $|\pm 3/2\rangle_{GS} \leftrightarrow |\pm 3/2\rangle_{ES}$ transitions and labelled $A_1$ and $A_2$, respectively. The energy separation between the two transitions is about 1 GHz, which is mainly determined by the ES zero field splitting[22]. It was recently shown that all optical transitions are fully spin conserving, which provides several pathways for spin–photon entanglement generation[9,25]. However, spin-flip processes can still be mediated by non-radiative intersystem crossing involving metastable states, which allows e.g. for deterministic spin state initialisation[22,26]. Note that we do not consider the system's second ES, V1', whose fluorescence at 858 nm is strongly reduced at cryogenic temperatures due to ultrafast relaxation to the V1 state[20]. Thus, and as shown

in Fig. 1c, off-resonant excitation results in photoluminescence in the V1 ZPL (with 8% Debye–Waller factor[27]) or the associated phonon side band (PSB) with 6 ns lifetime[20].

**High-resolution ZPL emission spectrum**. Single h-$V_{Si}$ centres are addressed via confocal microscopy (see "Methods"). Figure 1d shows the fine structure of the V1 ZPL emission, recorded at a temperature of $T = 5.0$ K. To this end, we use continuous-wave off-resonant excitation at 730 nm with an optical power of about 0.15 mW, which is almost 3 times higher than the saturation power[20]. As a high-resolution spectrometer, we use a home-built tunable Fabry–Pérot filter cavity with a linewidth of $29 \pm 2$ MHz. By integrating the emission spectrum over 20 min, we clearly resolve the two optical lines corresponding to the spin-conserving $A_1$ and $A_2$ transitions. Surprisingly, even under strong off-resonant excitation, the linewidths remain very close to the lifetime limit (~27 MHz). Taking into account the finite spectral resolution of the filter cavity ($29 \pm 2$ MHz) via deconvolution of two Lorentzians, we infer emission linewidths of $57 \pm 6$ and $48 \pm 6$ MHz for the $A_1$ and $A_2$ transitions, respectively.

**Single-photon emission purity**. Aside from superior spectral purity and robustness, perfect two-photon interference crucially requires the interfering photons to be in pure single-photon states, i.e. not degraded by spurious multi-photon contributions[28]. To evaluate the quality of our quantum light source, we use pulsed off-resonant excitation at 780 nm (PicoQuant LDH-P-C-780, pulse length: 56 ps, repetition rate: 20.5 MHz) and record the second-order autocorrelation function $g^2(\tau)$ of ZPL photons in a standard Hanbury Brown and Twiss arrangement[29]. As shown in Fig. 1e, we observe a strong antibunching at zero time delay, clearly evidencing that the defect is a single-photon source ($g^2(\tau = 0) < 0.5$). Figure 1f shows the recorded value of $g^2(\tau = 0)$ as a function of the laser pulse energy above saturation (4 pJ, see Supplementary Note 1). The increase of $g^2(\tau = 0)$ is mainly due to noise from background fluorescence below 6 pJ (see Supplementary Note 3). Above 6 pJ, the increased probability to induce two optical excitations during one laser pulse degrades $g^2(\tau = 0)$. However, under all conditions, we observe $g^2(\tau = 0) < 0.25$ without any background subtraction, which underlines the h-$V_{Si}$ centre's single-photon emission quality.

**Indistinguishable photon generation**. To prove that the h-$V_{Si}$ centre generates streams of indistinguishable photons in the $A_1$ and $A_2$ transitions, we perform a Hong–Ou–Mandel (HOM) interference[30] experiment on two consecutively emitted photons[31]. Figure 2a shows the related set-up schematically. We use the off-resonant picosecond laser to excite the single defect twice with an interval of $\delta t = 48.7$ ns, and we repeat this sequence every $10 \cdot \delta t = 487$ ns. The laser pulse energy is set to 5.5 pJ, corresponding to about 74% excitation probability per pulse and the detected ZPL photon count is about $10^{-4}$ per pulse (see Supplementary Note 1). The ZPL emission is coupled into a single-mode fibre-based unbalanced Mach–Zehnder interferometer with a path travel time difference of $\delta t = 48.7$ ns. The interferometer's outputs are connected to superconducting nanowire single-photon detectors (SNSPDs; Photon Spot Inc.) with sub-ns time resolution. We record the time differences between both SNSPD detection events using a time tagger (Swabian Instruments). The timing jitter of our detection system is about 0.4 ns.

Let us assume now that two consecutive ZPL photons enter the interferometer during one experimental sequence (i.e. no non-radiative intersystem crossing occurred). The most interesting case occurs when the early and the late photons take the long and short interferometer paths, respectively, and thus arrive simultaneously

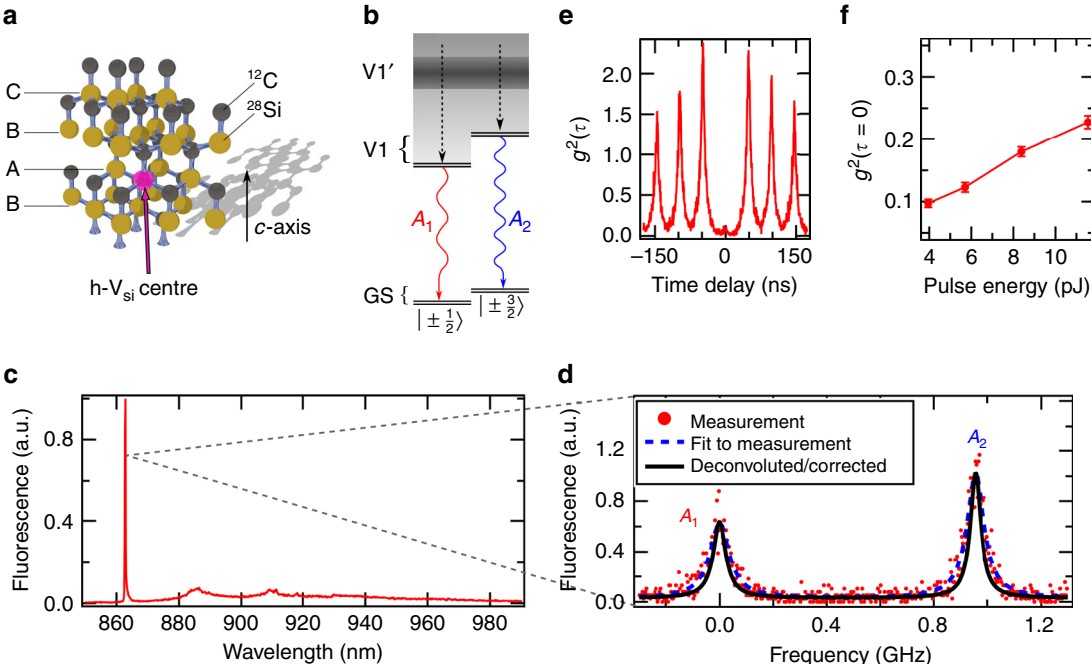

**Fig. 1 Optical properties of single silicon vacancies h-V_Si.** **a** Crystallographic structure of 4H-SiC and position of h-V_Si centre (highlighted by a pink sphere symbolising a missing Si atom). **b** Level structure of the h-V_Si centre at zero magnetic field. Ground state (GS) and V1 excited state show degenerate sublevels $m_S = \pm\frac{1}{2}$ and $m_S = \pm\frac{3}{2}$. Optical transitions between V1 and GS are spin conserving and associated with the transitions $A_1$ and $A_2$. Emission from the second excited state, V1′, is not observed owing to ultrafast relaxation (dashed arrows). **c** Single h-V_Si centre emission spectrum under off-resonant excitation. **d** ZPL emission fine structure recorded over 20 min. The red dots are raw data. The two emission lines associated with $A_1$ and $A_2$ transitions are clearly resolved. The blue dashed line is a Lorentzian fit to the raw data, giving linewidths (FWHM) of 86 ± 6 and 77 ± 6 MHz for the $A_1$ and $A_2$ lines, respectively. The black line is data after deconvolution correction for the finite linewidth of the scanning Fabry–Pérot cavity (Lorentzian FWHM of 29 ± 2 MHz), the resulting real emission linewidths are 57 ± 6 and 48 ± 6 MHz, respectively, which is very close to the Fourier transform limit. **e** Second-order autocorrelation function recorded for a single h-V_Si centre under pulsed off-resonant excitation (pulse energy: 5.7 pJ). We observe $g^2(\tau = 0) = 0.12 \pm 0.01$, clearly indicating single-photon emission. **f** $g^2(\tau = 0)$ as a function of laser pulse energy. The line is a guide to the eye. Error bars correspond to one standard error.

at the output beam-splitter (BS₂) through different inputs. Provided that both photons are indistinguishable, HOM interference occurs and they must leave BS₂ as a pair through the same output port[30]. Experimentally, this is measured through the reduction of coincidence events at zero time delay between both SNSPDs. Note that the normalised coincidence rate reduction is a direct measure of the photon indistinguishability. The normalisation process is actually implemented automatically in our set-up. Namely, 25% of the successfully created photon pairs take wrong opposite paths, i.e. the early and the late photons take the short and long interferometer paths, respectively. In this case, they arrive at BS₂ with a time difference of $\pm 2 \cdot \delta t$. As this delay exceeds the ES lifetime (6 ns), no interference is observed. Similarly, the remaining 50% of the paired photons choose identical interferometer paths and arrive at BS₂ without interference with a time difference of $\pm \delta t$. Thus we expect that the five detection rates associated with the different coincidence time differences $A_{-2 \cdot \delta t}$, $A_{-\delta t}$, $A_0$, $A_{+\delta t}$ and $A_{+2 \cdot \delta t}$ show the well-known ratio of 1:2:0:2:1[31].

Figure 2b shows typical raw data for the HOM experiment. The central peak at $\tau = 0$ ns is strongly suppressed compared to the side peaks at $\tau = \pm \delta t$ and $\tau = \pm 2 \cdot \delta t$. Each peak exhibits an exponential decay of 6 ns corresponding to the V1 ES lifetime. Following the analysis by Santori et al.[31], we determine the raw HOM interference visibility $V_0 = 1 - 2 \cdot \frac{A_0}{A_{-\delta t} + A_{+\delta t}} = 0.69 \pm 0.02$. Since this value greatly exceeds $2 \cdot g^{(2)}(\tau = 0)$, generation of two-photon entanglement can be straightforwardly implemented[32–34]. The HOM interference visibility is comparable to those reported for two consecutive photons from quantum dots with off-resonant excitation (0.53–0.86)[35–37].

**Spin-controlled distinguishable photon generation.** As mentioned earlier (see Fig. 1b, d), optical transitions are spin conserving and intimately linked to spin sublevels, i.e. $|\pm 1/2\rangle_{GS} \leftrightarrow |\pm 1/2\rangle_{ES}$ levels are connected via the $A_1$ optical transition and $|\pm 3/2\rangle_{GS} \leftrightarrow |\pm 3/2\rangle_{ES}$ via $A_2$. Consequently, without intersystem crossing, repeated off-resonant excitation leads to streams of consecutively emitted indistinguishable photons in the $A_1$ or photons in the $A_2$ transition, respectively. As one of the system's key assets, we highlight that the h-V_Si centre in SiC possesses a GS spin quartet with a coherence time that exceeds the ES lifetime by at least five orders of magnitude[20,22]. This provides a direct control mechanism of the colour of consecutively emitted photons, e.g. by flipping the GS spin from the subspace $|\pm 1/2\rangle_{GS}$ to $|\pm 3/2\rangle_{GS}$, the first and second photon will be in $A_1$ and $A_2$, respectively. Figure 3a shows the experimental sequence. The four GS spin levels are non-degenerate due to the external magnetic field. After the first excitation, we apply a short radiofrequency (RF) wave pulse at a frequency of 30.3 MHz. Note that, owing to the high RF pulse power and the short pulse duration, all GS spin transitions are driven simultaneously. However, our independent resonant excitation measurements and theoretical modelling show that spin driving between the $|\pm 1/2\rangle_{GS}$ and $|\pm 3/2\rangle_{GS}$ subspaces still reaches fidelities of about 70% (for more details, see Supplementary Note 4). Figure 3b shows the resulting HOM interference pattern at RF power of 30 dBm and a pulse length of 19 ns, which corresponds to a $\pi/2$-pulse (see Supplementary Note 4). The coincidence peak at zero time delay reappears clearly, due to the emission of distinguishable photons. In addition, as shown in Fig. 3c, the interference pattern around zero

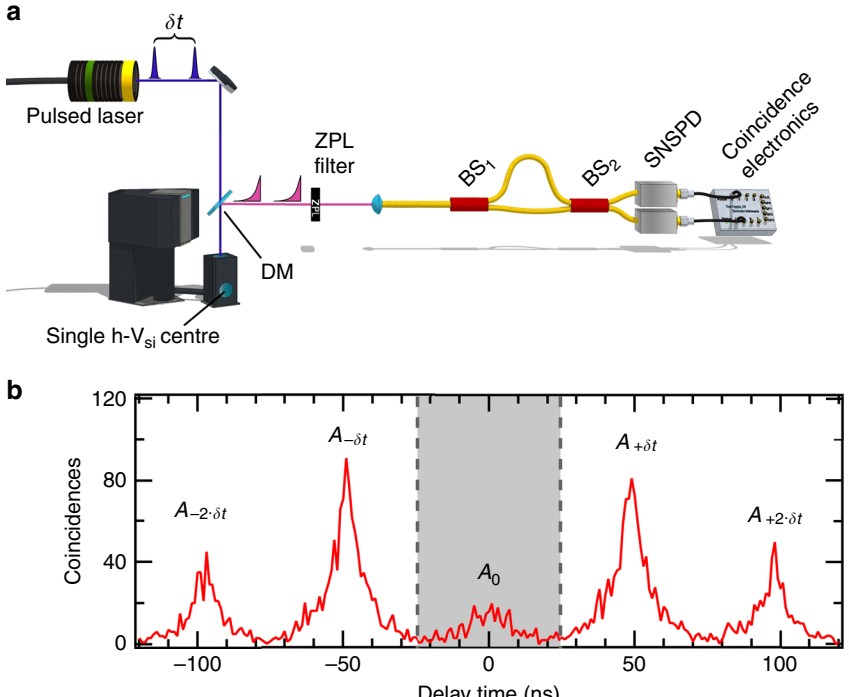

**Fig. 2 Hong–Ou–Mandel interference for photon indistinguishability study. a** Schematic set-up for HOM interference with two photons from a single h-$V_{Si}$ centre in SiC. Two laser pulses excite the h-$V_{Si}$ centre with a time delay $\delta t = 48.7$ ns. Consecutively emitted ZPL photons are sent to an unbalanced Mach–Zehnder interferometer. SNSPD and coincidence electronics are used to record the two-photon statistics at the output. **b** Two-photon coincidence counts as a function of the detection time delay between both SNSPDs. The strongly suppressed peak at zero time delay witnesses Hong–Ou–Mandel interference. The grey area between dashed lines symbolises the integration time window that is used for evaluating the interference contrast $V_0 = 0.69 \pm 0.02$. A two-photon coincidence occurs approximately every 84 s. The results shown are raw data without any correction.

time delay shows the expected modulation with a frequency of ≈1 GHz, matching the frequency difference between $A_1$ and $A_2$ optical transitions[38]. We note that the high RF pulse power causes significant optical linewidth broadening due to thermal heating, limiting the maximum HOM visibility to $0.56 \pm 0.04$ and $V_{max} = 0.73 \pm 0.05$ after correcting for experimental imperfections (as discussed later and in Supplementary Note 4). By normalising the recorded and corrected visibility $V_{measured}$ to $V_{max}$, we obtain a corrected HOM visibility of $V_{norm,\pi/2} = \frac{V_{measured}}{V_{max}} = 0.65 \pm 0.07$, which matches the expectation for a $\pi/2$-pulse (for details, see Supplementary Note 4). To prove that our spin-to-photon interface is indeed based on coherent spin manipulation, we repeat the experiment with two additional RF pulse lengths of 10 and 29 ns (corresponding to a $\pi/4$-pulse and $3\pi/4$-pulse, respectively). The observed visibilities are $V_{norm,\pi/4} = 0.90 \pm 0.09$ and $V_{norm,3\pi/4} = 0.47 \pm 0.17$, respectively. Data for the three measurements above is shown in Fig. 3d. The data match the theoretical model very precisely, which is based on independently measured Rabi oscillations (see Supplementary Note 4). Consequently, the results corroborate that the photonic emission of the h-$V_{Si}$ centre can be controlled by coherent spin manipulation, which is crucial for remote entanglement generation[5–8,39,40].

**Temperature stability of spin–photonic properties.** In the last step, we provide a deeper insight into the system's temperature-dependent spin–phonon–photon physics. For this, we consider the visibility reduction in the raw data of the HOM experiment without RF pulses. To explain deviations from non-unity visibility, we consider six detrimental factors: (i) pure dephasing in the ES due to phonon scattering[41], (ii) spectral diffusion due to local charge redistribution[42], (iii) two-photon emission during one

laser pulse[31,43], (iv) background counts, e.g. laser breakthrough, bulk and surface fluorescence, and Raman scattering, (v) interferometer imperfections, e.g. non-unity fringe contrast and unbalanced beam splitter transmissivity and reflectivity[31], and (vi) the photon arrival timing jitter at $BS_2$, e.g. due to finite laser pulse duration[44] as well as timing jitter of the laser. We infer that factors (v) and (vi) amount to 1% and 0.8% contrast reduction, respectively (see Supplementary Notes 2 and 3). Experimentally, we minimise the fast components of factors (iii) and (iv) by time-gating the coincidence detection, i.e. detector clicks are only considered posterior to a delay time $t_{Start}$ (see "Methods"). Generally, we find that the visibility saturates for $t_{Start} > 1.5$ ns, indicating that laser-induced noise is efficiently filtered out at this point. To infer the contribution of pure dephasing and spectral diffusion, we extend the analysis by Thoma et al.[41] and include coincidence window gating with start and stop delays $t_{Start}$ and $t_{Stop}$, respectively (for more details, see Supplementary Note 5). This provides us with an additional tuning knob, as we can now actively control the time window $\Delta t = t_{Stop} - t_{Start}$ in which we give pure dephasing (single-phonon scattering in the ES) a chance to degrade the fringe visibility in the HOM experiments (see Fig. 4a). As an example, Fig. 4b shows the HOM pattern at $T = 5.0$ K for $t_{Start} = 3.5$ ns and a short time window $\Delta t = 4$ ns. For those settings, we obtain an uncorrected raw visibility of $V_{0,gated} = 0.85 \pm 0.04$. For arbitrary time gating settings, we find an analytic expression for the expected HOM visibility:

$$V = \frac{1}{(1-e^{-\Gamma \cdot \Delta t})^2}\left[\frac{\Gamma}{\Gamma+\gamma} + \frac{\Gamma}{\Gamma-\gamma}e^{-2\Gamma \cdot \Delta t} - \frac{2\Gamma^2}{\Gamma^2-\gamma^2}e^{-(\Gamma+\gamma)\Delta t}\right]. \quad (1)$$

Here $\Gamma = \frac{1}{6\,ns}$ is the inverse ES lifetime, and $\gamma = \Gamma'_0 [1 - e^{-(\delta t/\tau_c)^2}] + 2\gamma'$, with $\Gamma'_0$ being the amplitude of spectral

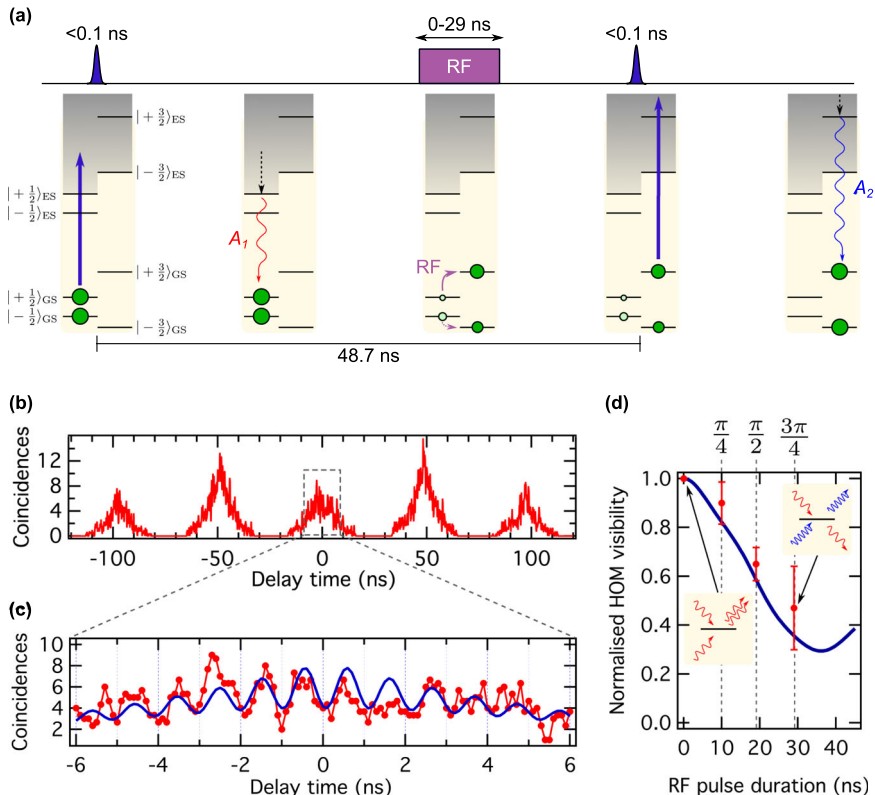

**Fig. 3 Distinguishable photon generation by spin control. a** Schematics for spin-controlled generation of distinguishable photons from a single h-$V_{Si}$ centre. The sketch shows the realisation when the system is initially in the $m_S = \pm\frac{1}{2}$ subspace. The first laser excitation results in a red photon ($A_1$ line). A subsequent RF pulse with variable duration transfers population (partially) from $m_S = \pm\frac{1}{2}$ to $m_S = \pm\frac{3}{2}$. The second excitation results in a blue photon ($A_2$ line), which makes the two interfering photons maximally distinguishable, and projects the system into the $m_S = \pm\frac{3}{2}$ subspace. **b** Two-photon coincidence counts as a function of the delay time for a RF pulse duration of 19 ns (corresponding to a $\pi/2$-pulse). The coincidence peak at zero time delay reappears. Data are recorded at 0.1 ns timing resolution and averaged over three points to improve signal to noise. **c** Zoom-in of the HOM interference pattern revealing the fringe pattern with the expected modulation at 0.965 ± 0.006 GHz. Red dots are uncorrected data, and the blue line is the fit to the data (for details on the fit model, see Supplementary Note 7). **d** HOM contrast as a function of the RF pulse duration. Rabi-like oscillations are observed, demonstrating that coherent spin manipulation controls the degree of photon indistinguishability. Red dots are data and the blue line is the theoretical model considering independently measured Rabi oscillations (see Supplementary Note 4). Error bars represent one standard error.

diffusion, $\tau_c$ the associated time constant and $\gamma'$ the pure dephasing rate of the single emitter. To infer $\gamma$, the experimental data has to be corrected for experimental imperfections (interferometer, finite signal-to-noise ratio and photon arrival timing jitter). Figure 4c shows the corrected HOM visibility as a function of $\Delta t$ for fixed $t_{Start} = 3.5$ ns. To gain more insights of the system, the measurements are performed by adjusting the cryostat to three different temperatures $T = 5.0$, 5.9, and 6.8 K. By fitting each data set with the model in Eq. (1), we extract $\gamma$. We note that one contribution to spectral diffusion is thermal ionisation of nearby charge traps, usually occurring at several tens of Kelvin[45–47]. Thus, in our experiments, spectral diffusion is attributed to laser ionisation, which depends very weakly on temperature and occurs usually at a microsecond time scales[47]. Consequently, we have $\tau_c \gg \delta t$, such that HOM visibility reduction is essentially only limited by pure dephasing ($\gamma'$). This permits us to provide an upper limit for the pure dephasing rate $\gamma'_{max}$ (obtained from the HOM experiments), as well as the associated amplitude of spectral diffusion $\Gamma'_0$ (obtained from slow absorption linewidth measurements). The results of the temperature-dependent measurements are summarised in Fig. 4d. As expected, we find that $\Gamma'_0$ remains nearly constant over the measured temperature range. The increase of $\gamma'_{max}$ is described by vibronic interaction of the V1 and V1' ESs[27], which are separated by a relatively small energy gap of $\Delta E = 4.4$ meV. As we detail in Supplementary Note 6, at the experimental temperatures, single

acoustic phonon scattering processes cause pure dephasing with a rate given by

$$\gamma'_{max}(T) = A \cdot \frac{(\Delta E)^3}{e^{\Delta E/k_B \cdot T} - 1}. \tag{2}$$

Here $k_B$ is Boltzmann's constant and the prefactor $A$ describes the phonon interaction strength. By fitting the data, we obtain $A = 2\pi \cdot (237 \pm 12)$ MHz (meV)$^{-3}$. Interestingly, we find that pure dephasing for the h-$V_{Si}$ centre is comparatively low at cryogenic temperatures, e.g. considering photonic interference experiments without any time-gating, pure dephasing reduces the optical coherence time to half of the transform limit ($2\Gamma^{-1} = 12$ ns) only at $T_{crit} > 6.9$ K. This temperature is comparable to quantum dot single-photon sources[48] and conveniently surpassed with standard cryostat equipment. Below $T_{crit}$, the remaining small emission linewidth broadening is mainly due to slow spectral diffusion. Although this does not present a significant influence for HOM experiments, it may be even further suppressed with improved crystal growth and associated annealing procedures to reduce charge traps. Alternatively, electronic device structures might be promising to control the charge environment[12,49–51].

## Discussion

In summary, we have demonstrated spin-controlled generation of indistinguishable and distinguishable photons from a single h-$V_{Si}$

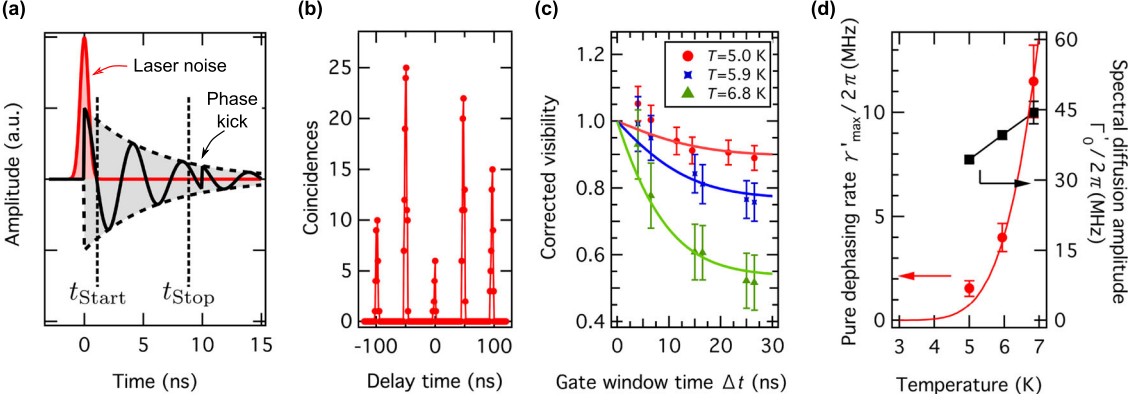

**Fig. 4 Pure dephasing of single photon state. a** Visualisation of the time-resolved HOM experiment (not to scale for better visualisation). The black solid line symbolises the single-photon wavefunction, while the black dashed line represents its envelope. For short times close to 0 ns, laser noise photons are observed (red pulse). This contribution is conveniently filtered out by only accepting detection events posterior to the start gate delay $t_{Start}$. In our visualisation, the photon wave function experiences a phase kick at a time delay of about 10 ns, due to phonon scattering in the V1 excited state. By having a variable stop gate time $t_{Stop}$, we effectively control the integration window $\Delta t = t_{Stop} - t_{Start}$. This allows us to infer the time scale of pure dephasing. **b** Uncorrected HOM pattern at $T = 5.0$ K for $t_{Start} = 3.5$ ns and $\Delta t = 4$ ns. Owing to the short time gating, the raw visibility increases to $V_0 = 0.85 \pm 0.04$, which matches the theoretical expectation ($0.80 \pm 0.01$) considering only experimental imperfections (see Supplementary Note 3). Dots are data and lines are visual guides. **c** Corrected HOM visibility at $t_{Start} = 3.5$ ns and varying $\Delta t$ and at three different temperatures. For $\Delta t \to 0$ ns, visibilities approach unity, as pure dephasing is completely suppressed. Dots are data (red circles: 5.0 K, blue cross marks: 5.9 K, and green triangles: 6.8 K), and lines with associated colours are fits for data at each temperature according to Eq. (1). **d** Temperature-dependent pure dephasing rate $\gamma'_{max}$ (red circles) and associated spectral diffusion amplitude $\Gamma'_0$ (black squares). The red line is the fit considering the model described with Eq. (2). The black line is a guide to the eye. Error bars in **c** and **d** correspond to one standard error.

centre in SiC. Despite performing our experiments under strong off-resonant excitation, pure dephasing, and spectral diffusion are exceptionally low, such that the quality of our raw data is sufficient for photonic entanglement generation[32–34]. Our semiconductor platform has thus demonstrated a level of spectral stability, which is on par with inversion-symmetry defects in diamond[13–16], quantum dots[36,41], and single molecules[34]. Furthermore, we showed that the system possesses an intimate spin–photon interface through which we deterministically tuned the degree of photon indistinguishability via coherent spin manipulation. Note that the system's millisecond spin coherence times[22] are sufficient for high-level spin–photon manipulation, such as remote entanglement generation[5–7] in a scalable quantum repeater network[8]. Implementing the related protocols necessitates optimised RF driving to reduce heating issues, which will suppress pure dephasing to negligible levels. To provide the necessary optical phase reference, resonant excitation is required, which requires several orders of magnitude lower optical powers, such that spectral diffusion will also be greatly suppressed[52]. In the perspective of quantum repeaters, long storage time quantum memories can be conveniently realised with recently demonstrated coupling to individual nuclear spins[22]. Implementation of state-of-the-art SiC photonic nanostructures would further increase light collection efficiency and ultra-compact chip-based integration[53], such that it will become realistic to use our system for generation of high-photon-number cluster states[54].

## Methods
**Sample preparation.** The 100-μm-thick 4H-$^{28}$Si$^{12}$C silicon carbide layer is grown by chemical vapour deposition on an n-type (0001) 4H-SiC substrate. The isotope purity is measured by secondary ion mass spectroscopy and inferred to be ∼99.85% for $^{28}$Si and ∼99.98% for $^{12}$C. Current–voltage measurements at room temperature shows that the layer is n-type with a free carrier concentration of ∼$6 \times 10^{13}$ cm$^{-3}$, which is close to the concentration of shallow nitrogen donors of ∼$3.5 \times 10^{13}$ cm$^{-3}$. Deep-level transient spectroscopic measurements show that the dominant electron trap in the layer is related to the carbon vacancy with a concentration in the mid $10^{12}$ cm$^{-3}$ range. Minority carrier lifetime mapping of the carrier shows a homogeneous carrier lifetime of ∼0.6 μs. We expect the real value to be twice as high, as an optical method with high injection was used[55]. Thus the density of all electron traps should be limited to the mid $10^{13}$ cm$^{-3}$ range[56]. Individually

addressable silicon vacancy centres were created through room temperature electron beam irradiation at 2 MeV with a fluence of $10^{12}$ cm$^{-2}$. Parasitic defects, such as carbon vacancies, interstitials, anti-sites, and their associated defects, are expected to be below mid $10^{12}$ cm$^{-3}$. Some interstitial-related defects were removed by subsequent annealing at 300 °C for 30 min.

To improve light extraction efficiency out of the high refractive index material ($n \approx 2.6$), we fabricate a solid immersion lens using a focussed ion beam milling machine (Helios NanoLab 650). The related surface contaminations and modifications are subsequently removed by wet and dry etching treatments[19].

**Experimental set-up.** All the experiments were performed at cryogenic temperatures in a Montana Instruments Cryostation. A home-built confocal microscope[20] was used for optical excitation and subsequent fluorescence detection of single silicon vacancies.

Continuous-wave off-resonant optical excitation of single silicon vacancy centres was performed with a 730 nm laser diode. Pulsed off-resonant excitation at 780 nm was performed using a picosecond laser diode (PicoQuant LDH-P-C-780). For resonant optical excitation at 861.7 nm, we used an external cavity tunable diode laser (Toptica DLC DL PRO 850). All lasers are spatial-mode cleaned by coupling to a single-mode fibre. For the picosecond laser, we employ a 780-nm bandpass filter after the fibre output to suppress Raman noise. Light is focussed onto the sample with a vacuum-compatible microscope objective (Zeiss EC Epiplan-Neofluar ×100, NA = 0.9).

Note that the used 4H-SiC sample was flipped to the side, i.e. by 90° compared to the c-axis, such that the polarisation of the excitation lasers is parallel to the c-axis ($E||c$), which allows to excite the V1 ES with maximum efficiency[20,57].

The fluorescence emission is collected by the same microscope objective and split by a dichroic mirror (Semrock Versa Chrome Edge) into two parts, i.e. ZPL (at 861.7 nm) and PSB (at 875–950 nm). PSB fluorescence is detected using a silicon avalanche photodiode (Excelitas SPCM-AQRH-W4).

ZPL emission is directed to a fully fibred Mach–Zehnder-type interferometer with a path length difference of ≈10 m. A fibre polarisation controller in the long interferometer arm allows to match the photon polarisation rotation experienced in both arms. Both interferometer outputs are directed to SNSPDs (PhotonSpot Inc.) with 80–85% detection efficiency and sub-Hz dark count rates.

In order to coherently manipulate GS spin populations, microwaves are applied through a 50-μm-thick copper wire spanned over the 4H-SiC sample in close proximity to the investigated h-$V_{Si}$ defect centre.

**Filter cavity for ZPL fine structure investigation.** To investigate the fine structure of the ZPL, we use a home-built fibre-coupled plano-concave Fabry–Pérot cavity. The cavity has the following specifications: free spectral range: 5.145 GHz; finesse: 177 ± 10; linewidth: 29 ± 2 MHz; input-fibre-to-output-fibre transmission: 80%. The cavity length is tunable via a piezoelectric actuator on which one of the two mirrors is glued. The cavity housing is made from Invar to ensure good

thermal stability. The typical drift of the cavity was measured to be on the order of $5\,kHz\,s^{-1}$.

Further details on how the cavity was used for resolving the fine structure of the ZPL emission are given in Supplementary Note 8.

**Time-gating scheme**. To filter out noise components induced by the off-resonant high-power pulsed laser excitation and to perform time-resolved HOM experiments for investigation of pure dephasing, a time-gated photon detection scheme is implemented by software. All photon click signals from the SNSPDs are time tagged by coincidence electronics (Swabian Instruments Time Tagger 20) and referenced to the laser pulse timing. Via software postprocessing, we then implement time-gating by validating only photon clicks between $t_{Start} \leq t \leq t_{Stop}$ (here $t = 0$ represents the earliest possible photon arrival time). Owing to the software implementation on the Time Tagger 20, the experimental data points with different settings of $(t_{Start}, t_{Stop})$ are obtained within a single measurement.

## Data availability
All data are available upon reasonable request from the corresponding authors.

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

## Acknowledgements

We warmly thank Christoph Becher, Jürgen Eschner, and Mete Atatüre for fruitful discussions regarding data analysis. We thank Andy Steinmann, Mario Hentschel, Jonas Meinel, Stephan Hirschmann, Ilja Gerhardt, Timo Görlitz, Matthias Widmann, Torsten Rendler, Yu-Chen Chen, Thomas Kornher, Kai-Mei Fu, Karsten Frenner, Photon Spot Inc., PicoQuant GmbH, and Swabian Instruments GmbH for technical help. N.M., C.B., R.N., I.G., E.H., D.L., M.J., M.N., D.D., V.V., R.K., R.S., J.W. and F.K. acknowledge support by the European Research Council (ERC) grant SMel, the European Commission Marie Curie ETN "QuSCo" (GA No 765267), the Max Planck Society, the Humboldt Foundation, and the German Research Foundation (SPP 1601). R.N. acknowledges support by the Carl-Zeiss-Stiftung. N.T.S. acknowledges the Swedish Research Council (grant No. VR 2016-04068). J.U.-H. thanks the Swedish Energy Agency (43611-1). N.T.S. and J.U.-H. thank the Knut and Alice Wallenberg Foundation (Grant No. KAW 2018.0071). A.G., N.T.S., and J.U.-H. thank the EU H2020 project QuanTELCO (Grant No. 862721). T.O. thanks the Japan Society for the Promotion of Science (Grant No. JSPS KAKENHI 17H01056 and 18H03770). A.G. acknowledges the National Excellence Program of Quantum-Coherent Materials Project (Hungarian NKFIH Grant No. KKP129866), the EU QuantERA Q-Magine Project (Grant No. 127889), the QuantERA Nanospin Project (Grant No. 127902) and the National Quantum Technology Program (Grant No. 2017-1.2.1-NKP-2017- 00001). A.G. and J.W. acknowledge the EU-FET Flagship on Quantum Technologies through the project ASTERIQS. J.W. and F.K. acknowledge the EU-FET Flagship on Quantum Technologies through the project QIA.

## Author contributions

N.M., R.N., J.W., and F.K. conceived and designed the experiment; N.M., I.G., E.H., D.L., and F.K. performed the experiment; C.B., M.J., V.V., R.K., and R.S. provided experimental assistance; N.M., P.U., G.T., A.G., J.W., and F.K. analysed the data; J.U.-H. and N.T.S. prepared and characterised SiC materials; T.O. performed electron beam irradiation; R.N. fabricated solid immersion lenses; N.M., I.G., and M.N. developed software for data acquisition and experimental control; N.M., C.B., D.L., D.D., V.V., P.U., G.T., A.G., J.W., and F.K. provided theoretical support. N.M., J.W., and F.K. wrote the manuscript. All authors provided helpful comments during the writing process.

## Competing interests

The authors declare no competing interests.
