## [Peer Review File · Nature Communications]

Reviewers' comments:

Reviewer #1 (Remarks to the Author):

This manuscript from N. Morioka et al. reports on the emission of indistinguishable photons from SiC defects. The authors show that single-photon emission can be efficiently controlled with a combination of laser and RF pulses, obtaining good experimental values for spin manipulation, g_2 and indistinguishability. The paper is well written, the results presented and discussed clearly. I think the topic and the quality of the paper are suitable for publication in Nature Communications. Nevertheless, a few minor issues must be addressed before publication.

- I could not find in the main text any comment on the statistics of these experiments. Have these experiments been performed on only one emitter? How many emitters have been investigated in total? How repeatable are the results?
- In the conclusion paragraph, the authors compare the spectral stability of SiC emitters with other platforms. It would be useful to have a similar comparison also for the other selling points of the papers, for example indistinguishability and single-photon purity (g_2).
- At line 88, the authors talk about the deconvoluted linewidth with high-power of excitation. From the text, it is not clear how the resonances are deconvoluted and respect to what?
- For some reason the quality of Fig.3 is poor. It needs to be improved.
- In Fig. 4c the left axis is missing.

Reviewer #2 (Remarks to the Author):

This manuscript presents experimental measurements of two photon interference from photons sequentially emitted from the zero phonon lines (ZPLs) of a single V1 silicon vacancy in 4H-SiC. By controlling the spin state of the defect, the degree of indistinguishability between photons can be varied, and beating between photons can be observed. The authors also perform theoretical calculations of how pure dephasing and spectral diffusion affect the results. The results indicate that the ZPL can produce highly indistinguishable photons even when exciting non-resonantly and even take a step toward generating spin-photon entanglement using the frequency of the photons. I find this work to be very high quality, quite impressive, and the first of its kind for this defect. I also find the detailed analysis of the effects of pure dephasing and spectral diffusion quite useful. This manuscript is of sufficient importance and interest to the solid state quantum information science community that it should be published in Nature Communications.

While I do support publication, I have one main concern that should be addressed, and a few lesser points to bring up. The main concern is about the count rate of photons into the ZPLs. For quantum networks or cluster state generation, this seems like a very important figure of merit, and I can't find anything in the main text and only hints in the supplementary information. In particular, the authors make comparisons to other emitters like defects in diamond, quantum dot, and single molecules in terms of spectral stability. Another important feature of some of these systems is that most of the photons are emitted into the ZPL and have high photon count rates. It seems important to know what percentage of photons are emitted into the ZPLs and what count rate this results in. This seems like a concern for silicon vacancies in SiC.

Lesser points:

1. The discussion of coincidence window gating and how it relates to pure dephasing is a little

confusing. My understanding is that this is used to better eliminate photons emitted during the laser pulse, but I feel like I'm not following the concept very well or what Fig. 4(c) is getting at. The supplementary information helped a little, but not too much.

2. Fig. 4(c) is missing the vertical axis label, which I believe should be the corrected HOM visibility.

3. The Rabi oscillations shown in Supplementary Figure 5(a,b) look a little strange. The authors comment on this, saying that a detailed theoretical model will be given later. I did find the model, but I couldn't find an explanation. Is it related to off-resonant driving of the transitions?

Answers to the Reviewers' comments

Reviewer #1 (Remarks to the Author):

This manuscript from N. Morioka et al. reports on the emission of indistinguishable photons from SiC defects. The authors show that single-photon emission can be efficiently controlled with a combination of laser and RF pulses, obtaining good experimental values for spin manipulation, g_2 and indistinguishability. The paper is well written, the results presented and discussed clearly. I think the topic and the quality of the paper are suitable for publication in Nature Communications. Nevertheless, a few minor issues must be addressed before publication.

We thank the Reviewer for the nice comments regarding the quality and relevance of our experimental results. We also thank the Reviewer for the suggestions made, which definitely helped us to further improve the manuscript.

- I could not find in the main text any comment on the statistics of these experiments. Have these experiments been performed on only one emitter? How many emitters have been investigated in total? How repeatable are the results?

We performed the experiments on a randomly chosen $h-V_{Si}$ centre.

The reason is that our previous measurements (cf. R. Nagy et al., Nat. Commun. **10**, 1954 (2019)) demonstrated that almost all $h-V_{Si}$ in the sample provide very similar spin-optical properties. We also mention that repeating all the shown HOM experiments on several colour centres would consume several months' worth of time, which is out of the scope of this experimental demonstration.

However, we are aware of the necessity of reproducibility in science. For this reason, we added another chapter in the Supplementary Information (Supplementary Note 8). Here, we have performed detailed optical (single-photon) studies on six defects. In detail, we measured the zero-phonon line fine structure in emission for six defects. Emission linewidths were found in the range of 46-89 MHz. As a comparison, the defect on which HOM experiments have been performed shows a linewidth of 53 MHz (averaged over both optical transitions).

We believe that those additional experimental data are sufficient for underlying the repeatability of the results.

- In the conclusion paragraph, the authors compare the spectral stability of SiC emitters with other platforms. It would be useful to have a similar comparison also for the other selling points of the papers, for example indistinguishability and single-photon purity (g_2).

We thank the Reviewer for the suggestions. We added a sentence in Results section to compare the two-photon interference visibility, *i.e.* indistinguishability, with quantum dot systems. Here, we limit the comparison study to experiments that used off-resonant excitation to have similar experimental conditions.

Text:

Since this value greatly exceeds $2 \cdot g^{(2)}(\tau = 0)$, generation of two-photon entanglement can be straightforwardly implemented^{29–31}. **The HOM interference visibility is comparable to those reported for two-consecutive photons from quantum dots with off-resonant excitation (53-86%) [O. Gazzano *et al.*, Nat. Commun. 4, 1425 (2013), M. Gschrey *et al.*, Nat. Commun. 6, 7662 (2015), N. Somaschi *et al.*, Nat. Photon. 10, 340 (2016)].**

In addition, regarding the comparison of single-photon purity, we agree with the Reviewer that the comparison is important. However, g_2 is strongly dependent on experimental parameters and setup alignment. For this reason, we would like to avoid adding a sentence to the manuscript not to raise confusion. Instead, we would like to provide a comparison in this letter to the Reviewer below:

Our pulsed g_2 value of 0.10 at saturation pulse energy P_{sat} (4 pJ) is comparable to those reported for quantum dots (0.13 [O. Gazzano *et al.*, Nat. Commun. 4, 1425 (2013)], 0.26 [R. S. Daveau *et al.*, Optica 4, 178 (2017)], <0.01 [M. Gschrey *et al.*, Nat. Commun. 6, 7662 (2015)], 0.03 [N. Somaschi *et al.*, Nat. Photon. 10, 340 (2016)]) and for other colour centres (0.09 for NV centres and 0.04 for SiV centres in diamond at unreported pulse energy [M. Leifgen *et al.*, New J. Phys 16, 023021 (2014)]). However, it is not straightforward to compare g_2 values with other studies because different experimental parameters including excitation methods (continuous-wave or pulsed), repetition rates of excitation laser, excitation strength, and spectral filtering methods can greatly affect the g_2 value.

Our analysis of signal-to-noise ratio suggests that the background-corrected g_2 value at P_{sat} is estimated to be 0.01 (please see Supplementary Note 3, Supplementary Figure 3). Therefore, we believe that the single-photon purity could be greatly improved by adopting a better narrow-bandwidth spectral filter in the detection path or improving the fabrication process of SIL to reduce the background fluorescence.

- At line 88, the authors talk about the deconvoluted linewidth with high-power of excitation. From the text, it is not clear how the resonances are deconvoluted and respect to what?

We agree with the Reviewer that this explanation is too short and might be confusing. The point is that we use a scanning Fabry-Pérot cavity as a high-resolution spectrometer to characterise the frequency width of the two resonant emission lines A_1 and A_2 . The cavity has a finite transmission linewidth of 29 MHz, which is comparable to the natural emission linewidth of the defects. Consequently, the emission linewidth that we observe in an experiment is a convolution product between the defect's real emission linewidth and the cavity linewidth. Both, the defect and the cavity show a Lorentzian frequency profile. The convolution of two Lorentzians with width a_1 and a_2 yields a new Lorentzian with a width of a_1+a_2 , which is the width that is observed in an experiment. Consequently, to infer the natural emission linewidth of the defect, we measure the linewidth in the experiment and subtract the linewidth of the cavity.

In order to clarify this procedure, we have modified the main text:

Text:

As a high-resolution spectrometer, we use a home-built tunable Fabry-Pérot filter cavity with a linewidth of **29 ± 1 MHz**. By integrating the emission spectrum over 20 minutes, we

clearly resolve the two optical lines corresponding to the spin-conserving A_1 and A_2 transitions. Surprisingly, even under strong off-resonant excitation, the linewidths remain very close to the lifetime limit (~ 27 MHz). **Taking into account the finite spectral resolution of the filter cavity (29 ± 1 MHz) via deconvolution of two Lorentzians, we infer emission linewidths of 57 ± 6 MHz and 48 ± 6 MHz for the A_1 and A_2 transition, respectively.**

Note that we have also updated Fig. 1(d) with much improved data quality, which was possible after a change of detectors and replacing a faulty fibre connector. In the Figure caption, the deconvolution correction is mentioned a second time.

Text:

The blue dashed line is a Lorentzian fit to the raw data, giving linewidths (FWHM) of 86 ± 6 MHz and 77 ± 6 MHz for the A_1 and A_2 lines, respectively. After deconvolution correction for the finite linewidth of the scanning Fabry P erot cavity (Lorentzian FWHM of 29 ± 1 MHz), the resulting real emission linewidths are 57 ± 6 MHz and 48 ± 6 MHz, respectively, which is very close to the Fourier transform limit.

In the updated linewidth measurements, we optimized the excitation laser power to 0.15 mW for better signal-to-noise ratio. This value is three times higher than the saturation power (0.05 mW). The description in the main text is updated accordingly.

Also, we have provided updated information in Methods and a more extensive explanation at the end of the new Supplementary Note 8.

- For some reason the quality of Fig.3 is poor. It needs to be improved.
We have improved the figure quality.

- In Fig. 4c the left axis is missing.
We have corrected the axis label.

Reviewer #2 (Remarks to the Author):

This manuscript presents experimental measurements of two photon interference from photons sequentially emitted from the zero phonon lines (ZPLs) of a single V1 silicon vacancy in 4H-SiC. By controlling the spin state of the defect, the degree of indistinguishability between photons can be varied, and beating between photons can be observed. The authors also perform theoretical calculations of how pure dephasing and spectral diffusion affect the results. The results indicate that the ZPL can produce highly indistinguishable photons even when exciting non-resonantly and even take a step toward generating spin-photon entanglement using the frequency of the photons. I find this work to be very high quality, quite impressive, and the first of its kind for this defect. I also find the detailed analysis of the effects of pure dephasing and spectral diffusion quite useful. This manuscript is of sufficient importance and interest to the solid state quantum information science community

that it should be published in Nature Communications.

We thank the Reviewer for reading our manuscript and giving valuable comments and suggesting the publication in Nature Communications.

While I do support publication, I have one main concern that should be addressed, and a few lesser points to bring up. The main concern is about the count rate of photons into the ZPLs. For quantum networks or cluster state generation, this seems like a very important figure of merit, and I can't find anything in the main text and only hints in the supplementary information. In particular, the authors make comparisons to other emitters like defects in diamond, quantum dot, and single molecules in terms of spectral stability. Another important feature of some of these systems is that most of the photons are emitted into the ZPL and have high photon count rates. It seems important to know what percentage of photons are emitted into the ZPLs and what count rate this results in. This seems like a concern for silicon vacancies in SiC.

The Reviewer is right that for some high-level quantum applications, the achievable amount of ZPL photons is important. However, in our opinion, it is not yet clear whether a very high natural ZPL emission is the optimum strategy. The reason is that today's standard methods for single-shot solid-state spin state readout usually rely on Phonon Side Band emission (and not ZPL emission). Therefore, it is quite important to identify a suitable quantum system that allows one to engineer its photonic properties on demand. Typically, this means that the system should be compatible with photonic nanostructures, and provide some degree of (high-speed) tunability, e.g. Stark shift tuning. Stark shift tuning of $h-V_{Si}$ centres has been recently shown (M. Rühl et al., *Nano Lett.* **20**, 658 (2020)), and similar experiments are currently undertaken by the group of Prof. Dr. Jelena Vuckovic (unpublished). In addition, the SiC nanophotonics is a maturing field, and it seems possible to engineer optical properties of $h-V_{Si}$ centres according to the target application (cf. D. Lukin et al., *Nat. Photonics* (2019). <https://doi.org/10.1038/s41566-019-0556-6>).

Coming back to our experimental data, in the Supplementary Note 1, we show that the detected number of ZPL photons per single off-resonant excitation pulse reaches values of $1.5 \cdot 10^{-4}$. Our experimental repetition rate is about 2 MHz, resulting in a detected ZPL photon rate of about 600 counts per second (two laser pulses per one repetition). For the

measurement shown in Fig. 2(b), the typical coincidence rate is about one two-fold coincidence event every 84 seconds. We mention this information now in the caption of Fig. 2.

Text:

The grey area between dashed lines symbolises the integration time window that is used for evaluating the interference contrast $V_0 = 0.69 \pm 0.02$. **A two-photon coincidence occurs approximately every 84 seconds.** The results shown are raw data without any correction.

To explain the count rates, we mention that the Debye-Waller factor of the $h\text{-}V_{\text{Si}}$ centre was previously reported to be close to 40%. However recent studies with better near-infrared sensitive spectrometers show that the Debye-Waller factor is rather 7-8% (cf. P. Udvarhelyi et al., arXiv2001.02459 [quant-ph]). This means that there is quite a potential to tune optical properties on demand, as opposed to other systems with >70% ZPL emission, where spectral shaping seems quite limited. It will thus be feasible to reach much improved count rates, while still maintaining the access to single-shot readout. We mention that very promising results for divacancy colour centres in 4H-SiC have been shown by the group of Prof. Awschalom (arXiv:2003:00042 [quant-ph]), notably, they improved the Debye-Waller factor from 5% to 75% via tuning the defect into the photonic crystal cavity resonance.

In addition, several basic hardware improvements could be implemented to gain at least one order of magnitude in count rates. For example, microscope objectives with improved AR coating for the 800-900 nm range are soon going to be available, promising a 2-3 times improved count rate. Additionally, high-level experiments should be performed on SILs carefully centred on top of pre-characterised defects. At the moment, SILs are randomly produced on the SiC substrate, meaning that defects will almost never be in the SIL focus. *E.g.*, we find that the SIL count rate enhancement in our experiments is typically 4 (compared to bulk), while theory suggests a 10-fold enhancement for a well-placed SIL. Additionally, all our fibre connectors are standard Thorlabs products, *i.e.* non-centred fibre cores, which leads to a typical connector loss of about 20%. Considering that our single photons typically pass through 3-4 fibre connectors, one could gain another factor of 2 when all fibres are spliced together.

An important point for future research will also be to identify the system's quantum efficiency, *e.g.* through controlled coupling to plasmons. The low stiffness of SiC (compared to diamond) could suggest that the quantum efficiency for near-infrared emitters may still be high, however experimental measurements will be required, and they are in the scope of another project.

Lesser points:

1. The discussion of coincidence window gating and how it relates to pure dephasing is a little confusing. My understanding is that this is used to better eliminate photons emitted during the laser pulse, but I feel like I'm not following the concept very well or what Fig. 4(c) is getting at. The supplementary information helped a little, but not too much.

The Reviewer is right that our time-gating scheme is somehow new in the field, as it is usually challenging to apply to quantum dot systems with very short lifetimes on the order of the detection system timing resolution.

In our experiments, the excited state lifetime is more than 10 times longer than our system's timing resolution. Thus, we can apply an additional filtering stage to increase HOM visibilities, and infer some phononic material parameters. This allows us to address two sources of HOM visibility reduction independently:

- a) Laser-related noise is filtered out by rejecting all coincidence events in the first 3.5 ns.
- b) Pure dephasing related noise is addressed/suppressed by rejecting photons that arrive at (very) late moments.

Fig. 4(a) intends to visualise the filtering strategy. t_{Start} is chosen late enough to not be bothered by laser noise (typ. $t_{\text{Start}} = 3.5$ ns). Then, t_{Stop} is varied in order to reduce the time window in which coincidence detection is allowed. As shown in Fig. 4(a), pure dephasing leads to a phase “kick” on the emitted photon wavefunction (in the plot, it happens at around 10 ns). If two photons were made to interfere – one with and one without a phase kick, HOM visibility is expected to be reduced due to reduced wavefunction overlap. At a fixed temperature, the phase “kick” probability per unit time is constant, thus for shorter time intervals $\Delta t = t_{\text{Stop}} - t_{\text{Start}}$, we expect less perturbed photon wavefunctions, and thus higher HOM contrast. This relationship is shown in Fig. 4(c) for 3 different temperatures.

Equation (1) in the main part addresses this filtering strategy, too. Considering that laser noise is efficiently filtered out, then the HOM visibility depends only on some material parameters (Γ and γ), as well as the time window length Δt , which determines *how long we give pure diffusion (single phonon scattering) a chance to interfere in our HOM measurement*.

To clarify this point, we have updated one sentence in the main text, just before equation (1):

Text:

This provides us with an additional tuning knob, as we can now actively control the time window $\Delta t = t_{\text{Stop}} - t_{\text{Start}}$ in which **we give pure dephasing (single-phonon scattering in the excited state) a chance to degrade the fringe visibility in the HOM experiments** (see Fig. 4(a)). As an example, Fig. 4(b) shows the HOM pattern at $T = 5.0$ K...

In addition, for better understanding of Fig. 4(a), we shaded the photon wave packet in gray.

2. Fig. 4(c) is missing the vertical axis label, which I believe should be the corrected HOM visibility.

We have corrected the axis label.

3. The Rabi oscillations shown in Supplementary Figure 5(a,b) look a little strange. The authors comment on this, saying that a detailed theoretical model will be given later. I did find the model, but I couldn't find an explanation. Is it related to off-resonant driving of the transitions?

The non-optimal Rabi-like oscillations are due to the high-power drive. The Rabi frequency for our experimental parameters is around 14 MHz, while the ground-state zero field splitting is only $2 \cdot 2.25$ MHz. Thus, the radiofrequency pulses show reduced spin-selectivity, *i.e.* all allowed spin transitions are driven simultaneously.

For the settings that we chose, the Rabi-like spin population oscillations in the subspaces $m_S = \pm \frac{1}{2}$ and $m_S = \pm \frac{3}{2}$ are shown in the Supplementary Figures 5(a)-(b). We found an excellent agreement between the measured data and the simulation. Having confirmed the spin population behaviour under high-power RF drive independently, allows us then to straightforwardly predict the HOM contrast under RF drive (which is the solid line in the Main Text Figure 4(d)).

To clarify this point, we have changed a part in the Main Text to read now:

After the first excitation, we apply a short radiofrequency wave (RF) pulse at a frequency of 30.3 MHz. **Note that due to the high RF pulse power and the short pulse duration, all ground state spin transitions are driven simultaneously. However, our independent resonant excitation measurements and theoretical modelling show that spin driving between the $|\pm 1/2\rangle_{GS}$ and $|\pm 3/2\rangle_{GS}$ subspaces still reaches fidelities of about 70% (for more details, see Supplementary Note 4).**

Additionally, we clarify this strategy in the Supplementary Note 4.

Text in the second paragraph:

...RF power of 30 dBm. **Due to the high pulse power, the Rabi frequency exceeds the zero-field splitting, resulting in an oscillation pattern that is different from a simple cosine function. However,** we still observe oscillations...

Changes to satisfy editorial policies

1. The meaning of the error bars in all figures is defined (one standard error) in their captions.
2. The description on the data availability is added at the end of the manuscript.

Other minor changes

For better understanding of the manuscript, we updated two sentences in the main text:

1. Regarding the meaning of the highly coherent ground state spin:
As one of the system's key assets, we highlight that the h - V_{Si} centre in SiC possesses a ground state spin **quartet with a coherence time that exceeds the excited state lifetime by at least five orders of magnitude**^{20,22}
2. Regarding the measurement at different temperatures:
To gain more insights in the system, the measurements are performed by adjusting the cryostat to three different temperatures $T = 5.0$ K, 5.9 K and 6.8 K. By fitting **each data set** with the model in Eq. (1), we extract γ .

REVIEWERS' COMMENTS:

Reviewer #1 (Remarks to the Author):

In this revised version of the manuscript N. Morioka et al. have addressed all my questions and comments. I appreciate the new details about the measurements and estimation of the linewidth, and about HOM interference visibility. The new part about the repeatability of their measurements makes the paper a bit more solid, although six exemplary emitters are still not sufficient. However, I acknowledge that this kind of measurements are time-consuming and hard to perform on large ensembles.

Therefore, the paper is now suitable for Nature Communications and I recommend publication.

Reviewer #2 (Remarks to the Author):

I have read the revised version of the manuscript. As stated in my earlier review, I think this work is of very high quality and quite significant in the field of solid state quantum information science community. My main concern expressed in my previous report is that the count rate of photons into the zero phonon line is not made very clear. The authors have addressed this to some extent in the current version by adding the two photon coincidence count rate in the caption of Figure 2. I don't see why the authors couldn't include in the main text the count rate into the ZPL and the Debye-Waller factor they included in their response, but I'm fairly satisfied. I'm fairly satisfied with their response to my other more minor points as well. I recommend publication in Nature Communications.

Answers to the Reviewers' comments

Reviewer #1 (Remarks to the Author):

In this revised version of the manuscript N. Morioka et al. have addressed all my questions and comments. I appreciate the new details about the measurements and estimation of the linewidth, and about HOM interference visibility. The new part about the repeatability of their measurements makes the paper a bit more solid, although six exemplary emitters are still not sufficient. However, I acknowledge that this kind of measurements are time-consuming and hard to perform on large ensembles.

Therefore, the paper is now suitable for Nature Communications and I recommend publication.

We wish to express our appreciation to the Reviewer for reading the revised manuscript and suggesting the publication in Nature Communication. We understand the reviewer's concern that six defects are not enough for revealing the whole statistics. However, please let us explain further: we have not picked out good defects for the repeatability study but we randomly chose six defects across the entire sample for the measurements. Therefore, we believe that the result of the six defects show a certain degree of reproducibility.

Reviewer #2 (Remarks to the Author):

I have read the revised version of the manuscript. As stated in my earlier review, I think this work is of very high quality and quite significant in the field of solid state quantum information science community. My main concern expressed in my previous report is that the count rate of photons into the zero phonon line is not made very clear. The authors have addressed this to some extent in the current version by adding the two photon coincidence count rate in the caption of Figure 2. I don't see why the authors couldn't include in the main text the count rate into the ZPL and the Debye-Waller factor they included in their response, but I'm fairly satisfied. I'm fairly satisfied with their response to my other more minor points as well. I recommend publication in Nature Communications.

We would like to appreciate the Reviewer for reviewing our manuscript and for recommendation for publication in Nature Communication. Regarding the count rate and the Debye-Waller factor, we thank the reviewer to raise this concern once again. To clarify them, we have added them in the main text as below:

Text (DWF):

Thus, and as shown in Fig. 1(c), off-resonant excitation results in photoluminescence in the V1 ZPL (**with 8% Debye-Waller factor²⁷**) or the associated phonon side band (PSB) with 6 ns lifetime²⁰.

Text (ZPL count rate)

The laser pulse energy is set to 5.5 pJ, corresponding to about 74% excitation probability per pulse **and the detected ZPL photon count is about 10^{-4} per pulse** (see Supplementary Note 1).